# Spatio-Temporal Changes of Land Surface Temperature and the Influencing Factors in the Tarim Basin, Northwest China

**Alim Abbas** [1,2,3], **Qing He** [2,3,*], **Lili Jin** [4], **Jinglong Li** [1,2,3], **Akida Salam** [5], **Bo Lu** [6] **and Yierpanjiang Yasheng** [7]

1    College of Resources and Environmental Science, Xinjiang University, Urumqi 830046, China; alimabbas@sina.cn (A.A.); jinglong1219@163.com (J.L.)
2    Institute of Desert Meteorology, China Meteorological Administration, Urumqi 830002, China
3    Taklimagan Desert Meteorlgy Field Experiment Station of CMA, Qiemo 841000, China
4    Department of Atmospheric Sciences, Yunnan University, Kunming 650500, China; jinlili1984@126.com
5    Kezhou Meteorological Administration, Kezhou 845350, China; akidasalam@sina.com
6    China National Climate Center, Beijing 100081, China; bolu@cma.gov.cn
7    Xinjiang Meteorological Observatory, Urumqi 830046, China; yepjys@sina.com
*    Correspondence: qinghe@idm.cn

**Abstract:** Land surface temperature (LST) is an important parameter that affects the water cycle, environmental changes, and energy balance at global and regional scales. Herein, a time series analysis was conducted to estimate the monthly, seasonal, and interannual variations in LST during 2001–2019 in the Tarim Basin, China. Based on Moderate Resolution Imaging Spectroradiometer (MODIS) LST, air temperature, air pressure, relative humidity, wind speed, precipitation, elevation, and land-cover type data, we analyzed the spatio-temporal change characteristics of LST and the influencing factors. High LSTs occurred in the desert and plains and low LSTs occurred in surrounding mountain regions. The highest LST was recorded in July (25.1 °C) and the lowest was in January (−9.5 °C). On a seasonal scale, LST decreased in the order: summer > spring > autumn > winter. Annual LST showed an increasing trend of 0.2 °C/10 a in the desert and mountain areas, while the plains indicated a decreasing trend. In spring and autumn, western regions were dominated by a downward trend, whereas in winter a downward trend occurred in eastern regions. In summer, areas covered by vegetation were dominated by a downward trend, and desert and bare lands were dominated by an upward trend. Random forest (RF) model analysis showed that elevation was the most significant influencing factor (22.1%), followed by mean air temperature (20.1%). Correlation analysis showed that the main climatic factors air temperature, relative humidity, and elevation have a good correlation with the LST. Land-cover type also affected LST; during February–December the lowest LST was observed for permanent glacier snow and the highest was observed in the desert. El Nino and La Nina greatly influenced the LST variations. The North Atlantic Oscillation and Pacific Decadal Oscillation indices were consistent with the mean LST anomaly, indicating their considerable influence on LST variations.

**Keywords:** land surface temperature; Tarim Basin; random forest; influencing factor

## 1. Introduction

Land surface temperature (LST) refers to the temperature of the soil, water, buildings, and the vegetation canopy on the land surface [1], is a key parameter for describing thermal conditions [2] and is a common research topic in local and global environmental studies [3]. LST plays an important role in a variety of scientific studies, such as those on hydrology, ecology, and global climate change [4]. LST can be obtained from ground observations, remote sensing data retrieval, and reanalysis data based on surface energy balance model estimations. Although ground observations have high accuracy and temporal resolution, their point-scale representativeness and the sparse distribution of meteorological stations are major limitations for their research and application at regional to global scales.

Satellite datasets have been widely used to examine long-term LST trends at regional and global scales. On regional scales, a positive trend center over Eastern Europe and a negative trend center over central Siberia were found during 2003–2010 by analysis using Moderate Resolution Imaging Spectroradiometer (MODIS) data [5]. Over Antarctica, MODIS and reanalysis data (ERA-5 and ERA-Interim) were used to estimate the LST trends during 2000–2018 [6], and a warming rate of 0.9 K/10 a was found. Over global deserts, the seasonal cycle and inter-annual variability of the LST were evaluated using MODIS LST and multiple sets of reanalysis data [7]. This research showed an overall LST increase at a rate of 0.3 °C/10 a based on MODIS data during 2002–2015. Global deserts occupy approximately 25% of Earth's land surface [8] and are projected to expand because of global warming [9]. In recent years, with intensified research on climate change and environmental and ecological issues, the spatial patterns of LST changes urgently need to be quantified through research and analysis. Especially in semi-arid areas, clarifying the changes in the LST spatio-temporal pattern has important practical application value for the rational development and utilization of watershed resources, construction of ecological agriculture, maintaining ecological balance, improving resource utilization, and ensuring the health of the ecological environment of the watershed [10]. The annual average LST in northwestern China has increased since 2000 [11], and this trend will lead to changes in the ecological environment of the region. The land use/cover type are important factors that affect the spatial distribution of LST, and research on the impact of land use/land cover change (LUCC) on LST can contribute to the understanding of the interaction between global climate change and terrestrial ecosystems [12]. LST is also an important parameter that controls surface humidity; therefore, research on the temporal and spatial distributions of LST has important guiding significance for agricultural production in the Tarim Basin (TB).

In recent years, studies have been conducted on the climate evolution of TB; however, these have mostly focused on changes in air temperature [13] and river runoff [14]. The TB has sufficient heat resources; however, LST in this region has an obvious spatio-temporal distribution that is affected by the topography, underlying surface, and vegetation coverage. Using the TB in Xinjiang, as the study area, MODIS data collected from 2001 to 2019 were used to figure out the characteristics and influencing factors for changes in LST. The main aims of this research were to (1) derive LSTs from MODIS data from 2001 to 2019 and (2) investigate the driving factors for LST changes and the changes in average annual temperature, relative humidity, precipitation, air pressure, and wind speed. The LUCC, El Nino, La Nina, North Atlantic Oscillation (NAO), and Pacific Decadal Oscillation (PDO) indices data were used to analyze the LST distribution, which could provide a new concept for monitoring the thermal environment of the study area. Furthermore, this study provides important information for understanding the land-surface-atmosphere energy exchanges in TB.

## 2. Materials and Methods

### 2.1. Study Area

The study area is located in the southern Xinjiang Province and the largest inland basin in China, more than $53 \times 10^4$ km$^2$, and is located in northwestern China (Figure 1) [15]. It is roughly 1500 km wide and 600 km in length, and the altitude ranges from 800 m to 1300 m. Minimal precipitation and high evaporation occur in the TB, and hence the basin has a typical continental arid climate. The Taklamakan Desert (TD) is located at the center of the TB [16] and is approximately 1000 km long and 400 km wide and with an area of approximately 337,600 km$^2$. The main land types are farmland and shrub areas, with areas of 36,580 km$^2$ and 26,940 km$^2$, respectively [17]. From 1961 to 2019, the annual mean air temperature in the TB is 10.6 °C, ranges from 9.3 °C to 11.9 °C, and have an upward trend with rates of 0.3 °C/10 a; Annual mean precipitation is 76.3 mm ranges from 32.8 mm to 153.8 mm and have an upward trend with rates of 6.9 mm/10 a. Because of extreme climate conditions, habitation areas account for approximately 10% of the total area [18]. The oases

in the study area were distributed in the areas surrounding the TD, where they lay along river systems [19].

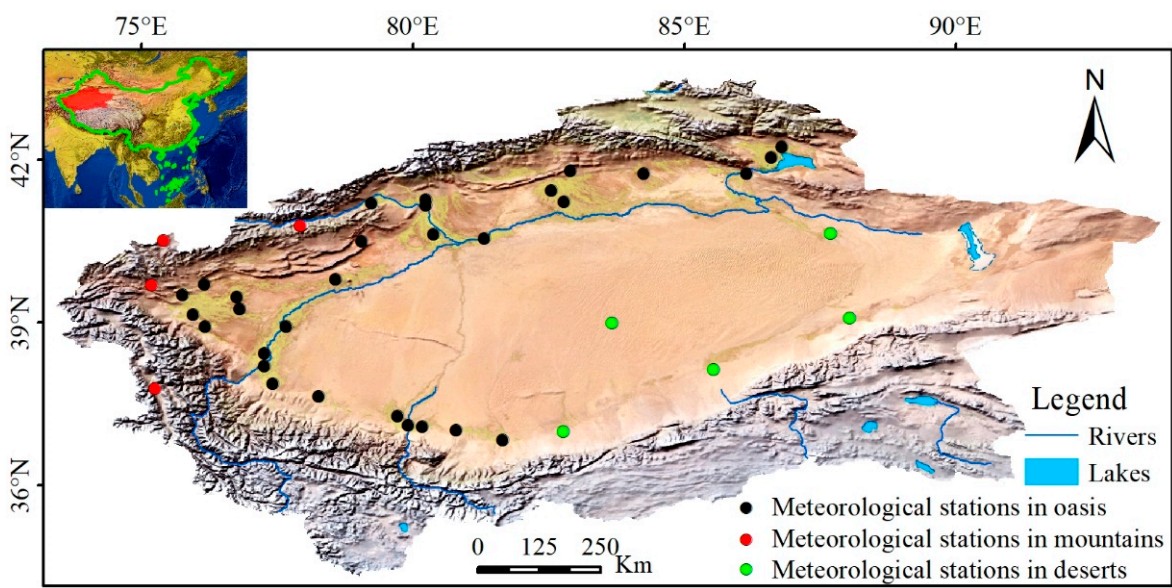

**Figure 1.** The territory of the study area. Located in northwestern China, lies between Tianshan Mountain, Kunlun, and Altun Mountain. The black, green, and red circles represent stations in oases, deserts, and mountainous terrain, respectively.

### 2.2. Data

#### 2.2.1. MODIS LST Data

In this study, the MODIS/Terra LST/Emissivity Daily L3 Global 1 km SIN Grid product data is used and downloaded from the NASA EarthData Search (see https://developers.google.com/earth-engine/datasets/catalog/MODIS_006_MOD11A1, last accessed on 11 March 2021), a Terra product was used, because it contains longer time series and greater time resolution (Terra daily, Aqua 8 days) data. MODIS/Terra passes over the Tarim Basin twice a day. The MOD11A1 product uses [20] (10.78–11.28 μm and 11.77–12.27 μm, respectively), downloaded by using the google earth engine platform. MODIS LST data is utilized in studies worldwide because of its daily global coverage [21,22]. Furthermore, many studies have shown its high estimation accuracy [23]. Hence, MODIS LST is the optimal choice for analyzing spatial and temporal LST variabilities. Daily MODIS 1 km LST data from January 2001 to December 2019 were used in our study, downloaded from the NASA EarthData Search (see https://developers.google.com/earth-engine/datasets/catalog/MODIS_006_MOD11A1, last accessed on 11 March 2021). Finally, the plot with ArcGIS 10.6 was performed.

#### 2.2.2. Meteorological Data

We used ground-based monthly maximum air temperature (Max AT), minimum air temperature (1.5 m ± 5 cm) (Min AT), mean air temperature (1.5 m) (Mean AT), air pressure (1.2 m) (AP), relative humidity (1.5 m) (RH), wind speed (10.8 m) (WS), precipitation (0.7 m ± 3 cm) (Pre), and elevation data from 39 weather stations during 2001–2019. A total of 4, 30, and 5 meteorological stations were selected to represent mountainous areas, oases, and desert areas, respectively [24]. For the verification of MODIS LST data in this study used the monthly LST situ-observation data (The sensing part and the surface body are half-buried in the soil) from 39 weather stations during 2001–2019. This study also used the annual average air temperature and precipitation data of 39 meteorological stations in the TB from 1961 to 2019 to introduce the characteristics of climate change of the study area. The locations of stations are presented in Figure 1. Meteorological data belong synoptic observation program and each series contains 8892 data (result part) and 59 data (study

area part). The data were provided by the Xinjiang Meteorological Administration and underwent strict quality control prior to being released.

Eight concomitant variables, Mean AT, Max AT, Min AT, AP, RH, WS, Pre, and elevation were considered. Within the RF model framework, concomitant variables were evaluated and the significance of concomitant variables was obtained.

### 2.2.3. Climate Index Data

NINO3.4 is the exceed 0.4 °C for 6 months running mean sea surface temperature anomaly in the region (5° S~5° N, 120° W~170° W) and has large variability on El Niño time scales [25]. The North Atlantic Oscillation (NAO) index is the indicator of the NAO and is the difference of normalized mean zonal sea level pressure between the Azores and Iceland [26]. The Pacific Decadal Oscillation (PDO) is a Decadal cycle of climate change in the Pacific Ocean and characterized by unusually warm or cold surface water temperatures in areas north of 20° N in the Pacific Ocean [27].

The NINO3.4, NAO, and PDO datasets that support the findings of this study are available (see https://psl.noaa.gov/gcos_wgsp/Timeseries/, last accessed on 28 May 2021). This data was copied from this website into a TXT file and then processed.

### 2.2.4. LUCC Data

Land-use data (Figure 2) were provided by the Data Center for Resources and Environmental Sciences, Chinese Academy of Sciences (RESDC) (see https://www.resdc.cn/data.aspx?DATAID=264, last accessed on 13 April 2021). The datasets are gridded at 30 × 30 m. The land-use types include 23 categories (Paddy field, Dryland, Forest land, Shrubwood, Sparse woods, Other woodland, High cover grassland, Medium covered grassland, Undercover grassland, Channel, Lakes, Reservoir pond, Permanent glacier snow, Bottomland, Urban land, Rural residential land, Other construction land, Desert, Gobi, Saline and alkaline land, Wetland, Bare land, and Bare rocky land).

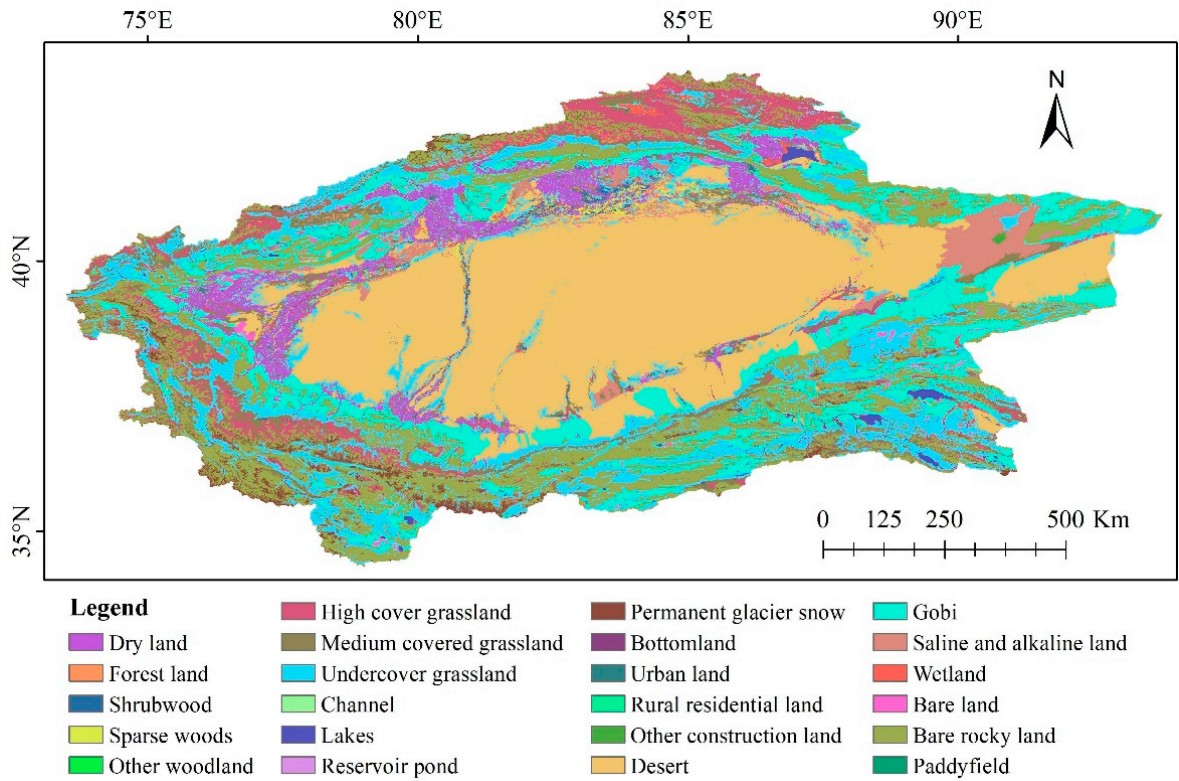

**Figure 2.** Land use in the study area in 2018.

### 2.3. Method

2.3.1. Sen's Slope Estimator

Sen's slope estimator is a common tool to for slope regression [28]. Sen's slope estimator is computed as

$$m_{ij} = \frac{Y_j - Y_i}{j - 1} \tag{1}$$

where, $i$ = 1 to $n - 1$, $j$ = 2 to $n$, $Y_j$ and $Y_i$ are data values at time $j$ and $i$ ($j > i$), respectively.

If in the time series there are $n$ values of $Y_j$, estimates of the slope will be $N = n\,(n - 2)/2$. The slope of the Sen Estimator is the mean slope of such slopes $N$ values. The Sen's slope is

$$m = \begin{array}{ll} m[\frac{n+1}{2}] & \textit{if n is odd} \\ \frac{1}{2}(m_{\frac{n}{2}} + m_{\frac{n+2}{2}}) & \textit{if n is even} \end{array} \tag{2}$$

Positive Sen's slope reveals an upward trend while negative suggests a downward trend.

2.3.2. The Mann–Kendall Test

The M-K test is widely used in the analysis of the climatological time series [29]. The M-K test statistic $S$ [30,31] is calculated as

$$S = \sum_{i=1}^{n-1} \sum_{j=i+1}^{n} \mathrm{sgn}(x_j - x_i) \tag{3}$$

where $n$ represents the number of data points, $x_i$ and $x_j$ are the data values in time series $i$ and $j$ ($j > i$), respectively and $\mathrm{sgn}(x_j - x_i)$ is the sign function as

$$\mathrm{sgn}(x_j - x_i) = \begin{cases} +1, & \textit{if } x_j - x_i > 0 \\ 0, & \textit{if } xj - xi = 0 \\ -1, & \textit{if } xj - xi < 0 \end{cases} \tag{4}$$

The variance is computed as

$$Var(S) = \frac{n(n-1)(2n+5) - \sum\limits_{i=1}^{m} t_i(t_i - 1)(2t_i + 5)}{18} \tag{5}$$

where $n$ represents the number of data points, $m$ is the number of tied groups and $t_i$ indicates the number of ties of extent $i$. The binding group is a set of sample data with the same value. Where the sample size $n > 10$, the standard normal test statistic $Z_S$ is computed using Equation (6):

$$Zs = \begin{cases} \frac{S-1}{\sqrt{Var(S)}}, & \textit{if } S > 0 \\ 0, & \textit{if } S = 0 \\ \frac{S+1}{\sqrt{Var(S)}}, & \textit{if } S < 0 \end{cases} \tag{6}$$

Positive values of $Z_S$ represent upward trends while negative $Z_S$ values show downward trends. In this study, significance levels $\alpha$ = 0.05 were used. At the 5% significance level and the null hypothesis of no trend is rejected if $|Z_S| > 1.96$.

2.3.3. Random Forest Model

The Random Forest (RF) model can process high-dimensional data and can be applied in gathering data. RF selection can be used to select partition variables of a minimized set for regression, and its output is the mean value of all decision-making trees [32]. The RF model uses multi-variate sorting for determining variables and clarifying their relative significance [33]. LST was a dependent variable.

## 3. Results

### 3.1. MODIS Product Verification

We matched the results of the MODIS LST product (MOD11A1) with the observed data from 39 weather stations in the TB for accuracy verification. Ground-based observations have a high precision and are extensively applied for LST verification [34]. As shown in Figure 3, we analyzed the monthly trends of ground-based LST and MODIS LST. The results show that the continuity of ground LST data and MODIS LST data were high, and they had a high coefficient of $R^2$ (0.98), the minimum LST difference between the two data is relatively low, while the maximum LST of MODIS is underestimated. This may be related to the pass-over time of MODIS Terra. This accuracy explains why MODIS LST products have increased in popularity and have been used by numerous scholars [35–37].

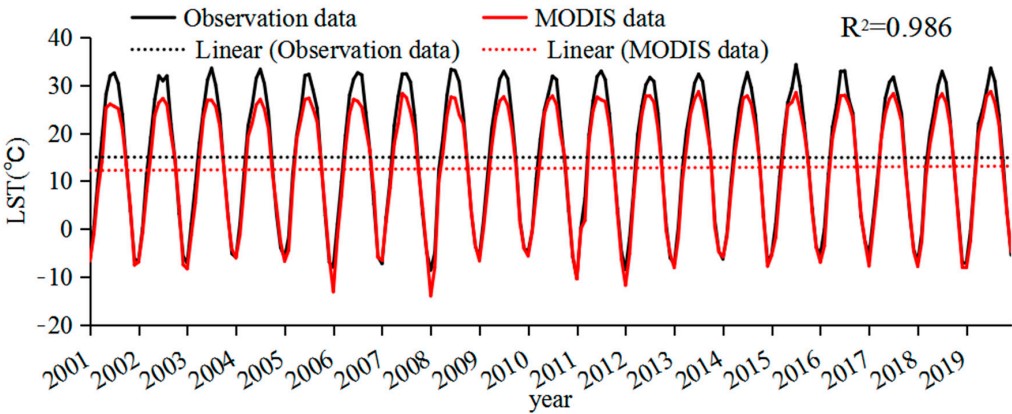

**Figure 3.** Trend comparison of MODIS LST data with ground−based observations.

### 3.2. LST Spatial Distribution Characteristics

Figure 4 illustrates the yearly variation of annual mean LST in the TB during 2001–2019. The multi-year average LST for all of the TB was 10.2 °C, and the annual mean LSTs were 10.3 °C, 11 °C, 11.3 °C, 11.5 °C, 10.5 °C, 12 °C, 10.7 °C, 11.4 °C, 11.2 °C, 10.3 °C, 11 °C, 10.3 °C, 11.9 °C, 10.7 °C, 11.9 °C, 11.8 °C, 11.6 °C, 10.8 °C, and 11.1 °C for 2001–2019, respectively. Most of the high LST values appeared in southwestern parts of the study area, and low LST values occurred near mountain regions. In addition, the spatial distributions of the average annual LST values showed similar patterns for all years.

Figure 5a shows the statistical changes of the LST spatial distribution in the TB. LST during 2001–2019 had a wavy trend and increased by 0.2 °C/10 a, although there was a gradual decline during 2006–2010 (12 °C to 10.2 °C). A low inflection point (10.2 °C) appeared in 2006, and then the LST increased. However, after 2013, LST considerably decreased. This may have been related to climate change throughout the TB. The center of the basin presented a high LST distribution, which may be because it is adjacent to the TD. Local dry weather and sparse rainfall contributed to the high LST.

Figure 6 shows the spatial distribution in average LST in different months. The results showed that the lowest average LST was observed during winter in January, at −9.5 °C in the surrounding mountainous region of the TB, which could have been caused by high latitudes and the mountainous highlands. Due to an increase in air temperature in February, the average LST increased by 1.8 °C compared to January. The same trend was seen in March. The highest winter LST (20.1 °C) occurred in March in the southern and some southwestern parts of the TB due to factors such as dry air, a lack of vegetation, and low latitudes. Due to the gradual warming of air temperatures, LST increased over the TB during spring. During spring, the lowest average LST (−33.9 °C) occurred in the southwestern, southern, and eastern parts of the desert, indicating a major role of high-altitude mountains. The LST distribution in May was the same as that in April, but with increased values. The highest average LST in spring occurred in May (35.1 °C) in the

southwestern and northeastern parts of the TB, strongly influenced by low altitudes, low latitudes, and the desert climate. A lack of moisture in this area may also have been the reason for the high LST. In June, July, and August, the effects of mountains and latitude were somewhat diminished in comparison with previous months, and a high average LST resulted from high air temperatures. In the first month of summer, the average LST was >40 °C in some parts of the TB, which directly correlated to the air temperature increase. A significant reduction in the slope of the LST was evident in September. A further decrease in the LST in the TB occurred leading up to winter. In October (i.e., autumn), LST ranged from −27.8 °C to 21.9 °C, indicating the continuation of the temperature decrease and the beginning of the cold season. This decrease continued in November and December. In November, the decrease in the LST was very rapid, such that the average LST in the basin was <0 °C except in a small area in the southwest. The lowest LST was observed over the entire basin in December, ranging from −37.5 °C to 4.7 °C, with a mean of −8.7 °C.

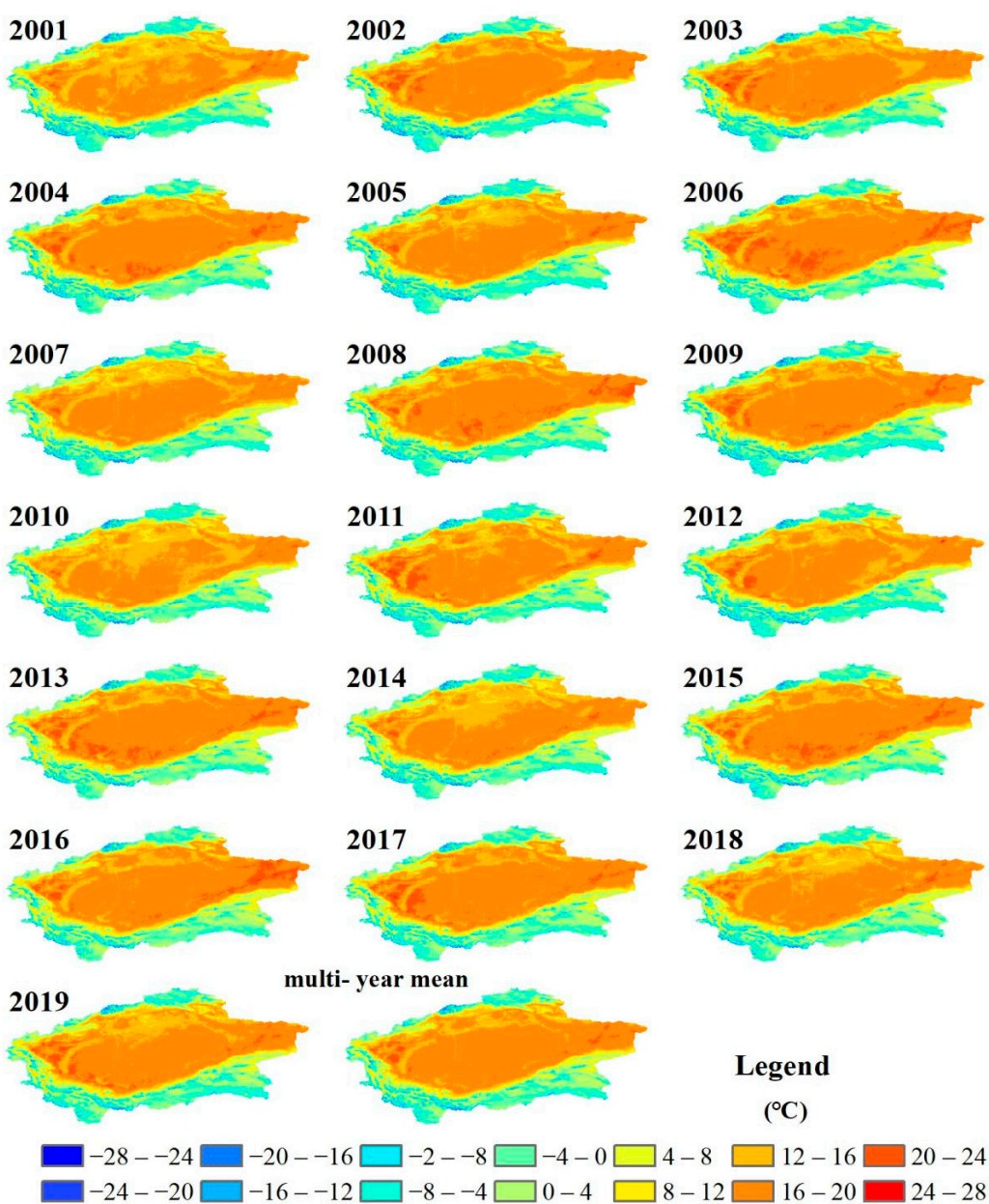

**Figure 4.** Spatial distributions of annual mean land surface temperatures (LSTs) across the Tarim Basin (TB) during 2001–2019.

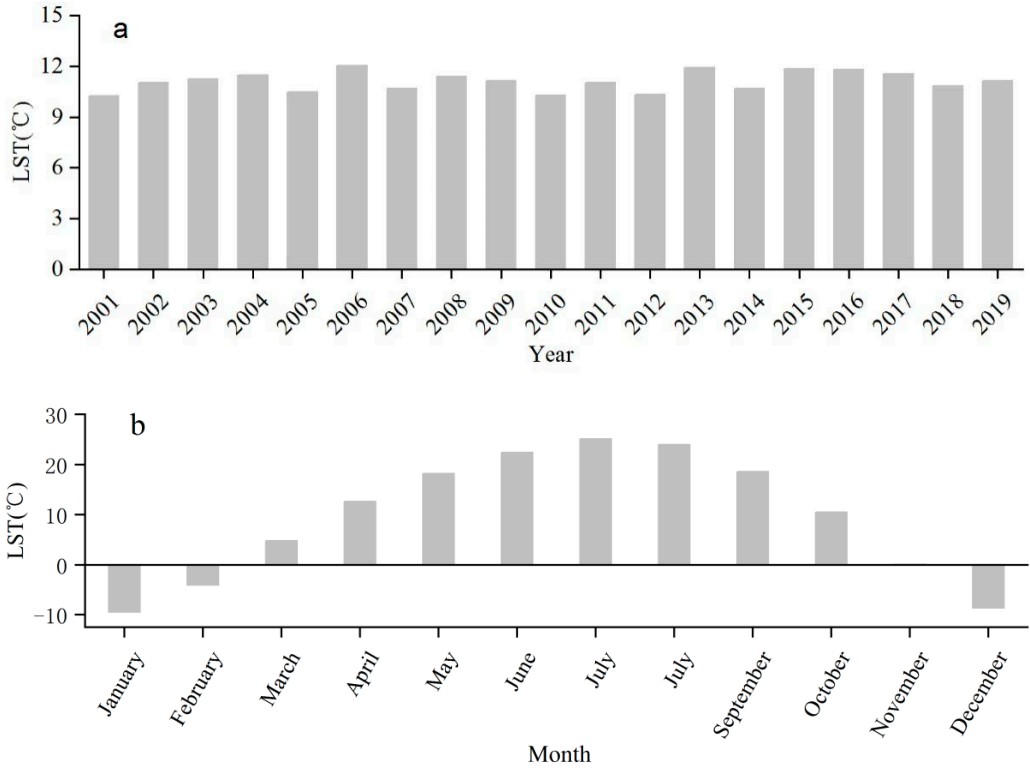

**Figure 5.** (**a**) Land surface temperature (LST) in the study region during 2001–2019 and (**b**) temporal distribution of the mean LST in different months during 2001–2019.

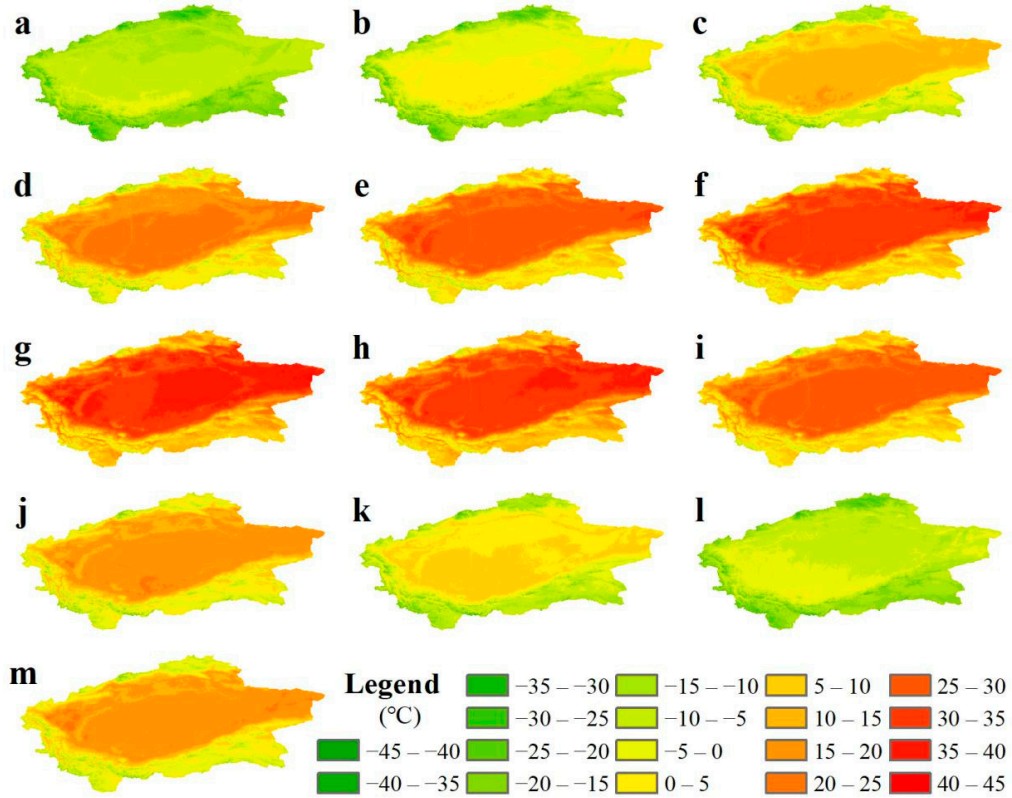

**Figure 6.** Spatial distribution of annual average land surface temperature (LST) in the Tarim Basin (TB) for different months: (**a**) January, (**b**) February, (**c**) March, (**d**) April, (**e**) May, (**f**) June, (**g**) July, (**h**) August, (**i**) September, (**j**) October, (**k**) November, (**l**) December, and (**m**) annually.

The highest average LST in the TB occurred in July (25.1 °C), and the lowest occurred in January (−9.5 °C). Figure 5b shows that the monthly change in LST in Xinjiang appeared as an obvious "unimodal" curve. The yearly variation in LST in the TB exhibited a "slow rise–significant sharp decline" trend (−9.5 °C to 25.1 °C to −8.7 °C).

Figure 6m shows the spatial distribution of the annual long-term mean LST over the TB. It is clear from the figure that the minimum LST (−24.9 °C) was observed in the surrounding mountains of the TD, which was caused by the high latitude and humidity. Meanwhile, the highest LST (23.3 °C) was observed in the southwestern parts of the basin, which, as mentioned earlier, may have been due to the dry atmosphere, low latitude, and lack of vegetation in the region. LST decreased from the center of the basin to the surrounding areas due to the altitude of the surrounding mountains.

### 3.3. Seasonal Time Series of LST

The year was divided into four seasons based on surface air temperature changes [38]. Figure 7 shows the seasonal spatial distribution of average LST during 2001–2019, and Figure 8 shows the maximum, minimum, and average seasonal values. Average LST in spring, summer, autumn, and winter ranged from −27.1 °C to 25.9 °C, from −16.8 °C to 41.3 °C, from −26.5 °C to 21.3 °C, and from −39.1 °C to 4.7 °C, respectively; the overall mean temperatures were 12.1 °C, 24 °C, 10 °C, and −8.3 °C, respectively (Figure 7). The distribution of average LST in spring, autumn, and winter was roughly similar, with high values in the southwestern parts of the TB and low values in the surrounding mountains. High air temperatures in summer resulted in an increase in average LST, which then steeply declined from summer to winter.

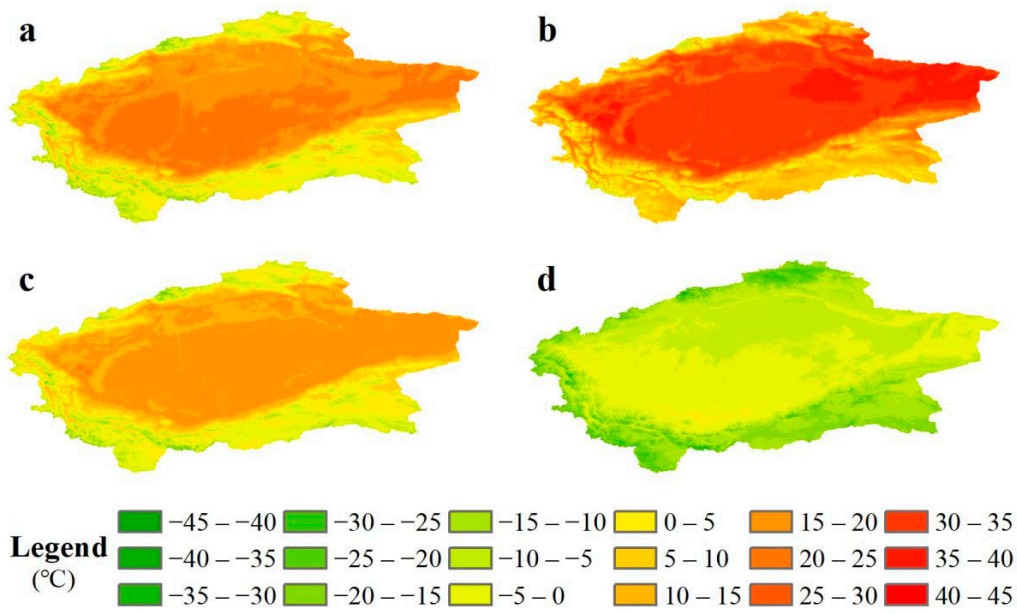

**Figure 7.** Seasonal mean land surface temperature (LST) distribution in the Tarim Basin (TB). (**a**) spring, (**b**) summer, (**c**) autumn, (**d**) winter.

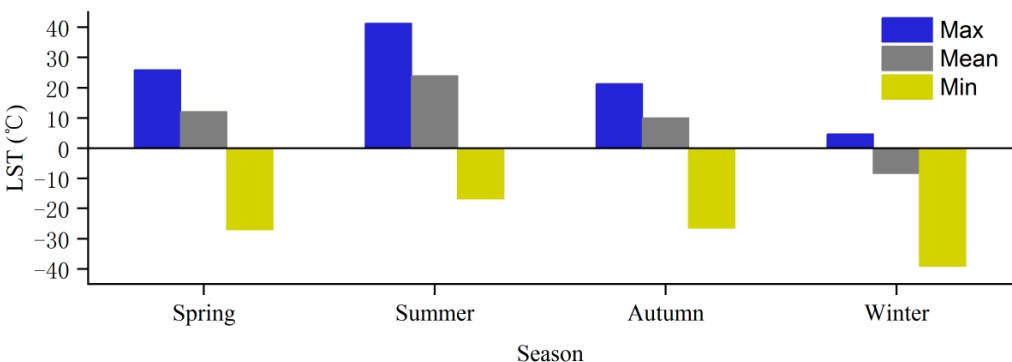

**Figure 8.** Distribution of the average land surface temperature (LST) in each season during 2001−2019.

### 3.4. LST Trends

By using M-K analysis, trends were calculated for each calendar month during 2001–2019. The upward and downward trends are indicated in red and green, respectively, in Figure 9, which shows LST trends for different months. In January, the annual LST rate was between −0.7 °C/10 a and 0.8 °C/10 a. The lowest rate (−0.7 °C/10 a) was observed in the eastern parts of the basin and the highest rate (0.8 °C/10 a) was evident in the western and southwestern parts of the basin. Meanwhile, in February, the LST tendency was between −1 °C/10 a and 1 °C/10 a. The lowest rate (−1 °C/10 a) occurred in the eastern and northwestern parts of the basin, and the highest rate (1 °C/10 a) occurred in the southern part of the basin. In March, the annual LST tendency was between −1 °C/10 a and 0.9 °C/10 a. The lowest rate (−1 °C/10 a) was observed in the southeastern and northern part of the basin, and the highest rate (1 °C/10 a) was observed in the eastern and southwestern and northeastern parts of the basin. LST in most parts of the study area showed an increasing trend. The LST tendency in April was between −0.6 °C/10 a and 0.6 °C/10 a. The lowest rate (−0.6 °C/10 a) was observed in the northern, western, and southern mountains; the highest rate (0.6 °C /10 a) was observed in the eastern part of the basin. Furthermore, significantly high-value centers existed in the northeast, and a low-value center was present to the southwest of the desert. The LST in most parts of the TB showed an increasing trend. In May, the LST tendency was between −0.7 °C/10 a and 0.5 °C/10 a. A low rate (−0.7 °C/10 a) was observed in most parts of the basin, and the highest rate (0.4 °C/10 a) was observed in the southern and southeastern parts of the basin. Significantly low-value centers occurred to the northeast of the desert, and LST in most parts of the study area showed a decreasing trend. In June, the LST tendency was between −0.6 °C/10 a and 0.6 °C/10 a. The lowest rate (−0.6 °C/10 a) occurred in the northern regions, and the highest rate (0.6 °C/10 a) was observed in desert. The LST in the basin was dominated by an increasing trend, and the LST in the surrounding mountains was dominated by a decreasing trend. In July, the LST tendency was between −0.7 °C and 0.5 °C/10 a. The lowest rate (−0.7 °C/10 a) was observed in the surrounding oases of the desert, and the highest rate (0.5 °C/10 a) was evident in the desert and northwestern parts of the basin. Most parts of the study area showed an increasing trend. In August, the LST tendency was between −0.8 °C/10 a and 0.7 °C/10 a. The lowest rate (−0.8 °C/10 a) was observed in the western part of the basin, and the highest rate (0.7 °C/10 a) was evident to the east of the basin. There was an obvious zonal distribution from low to high rates in the surrounding regions. LST in the eastern part of the basin demonstrated an obvious increasing trend, with an obvious decreasing trend in the west. In September, the LST tendency was between −0.5 °C/10 a and 0.5 °C/10 a, and the spatial distribution features were similar to those in August. In October, the LST tendency was between −0.5 °C/10 a and 0.7 °C/10 a. The lowest rate (−0.5 °C/10 a) was observed at the center of the desert, and the highest rate (0.7 °C/10 a) was observed in the surrounding regions of the basin. Most regions of the basin were dominated by LST rates from 0 °C/10 a to

−0.2 °C/10 a, while the surrounding regions were dominated by rates between 0 °C/10 a and 0.2 °C/10 a. In November, the LST rate was between −0.9 °C/10 a and 0.7 °C/10 a. The lowest rate (−0.9 °C/10 a) was observed in the northern part of the basin, and low values were observed in the southern parts of the basin. Most regions of the basin were dominated by LST rates from 0 °C/10 a to −0.2 °C/10 a. In December, the LST rate was between −0.6 °C/10 a and 0.9 °C/10 a. The lowest rate (−0.6 °C/10 a) was observed in the northeastern and southwestern parts of the basin, while the highest rate (0.9 °C/10 a) was observed in the northwestern parts of the basin. Most regions of the basin were dominated by rates from 0 °C/10 a to 0.2 °C/10 a.

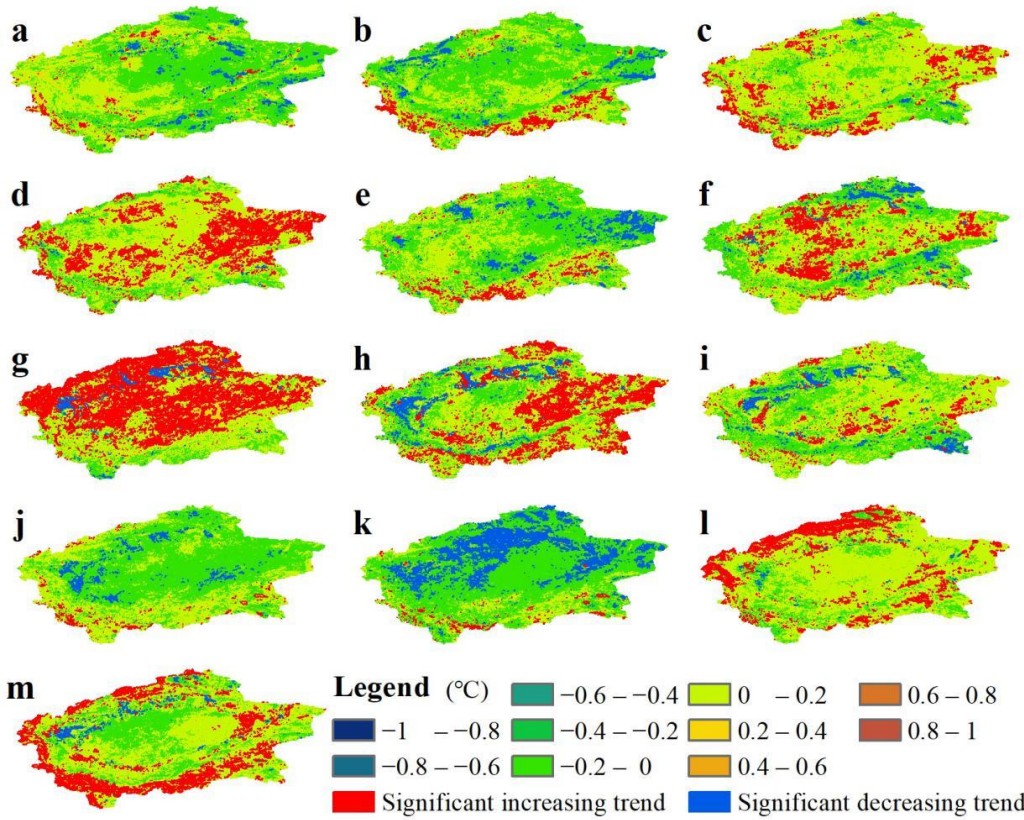

**Figure 9.** Land surface temperature (LST) trends for: (**a**) January, (**b**) February, (**c**) March, (**d**) April, (**e**) May, (**f**) June, (**g**) July, (**h**) August, (**i**) September, (**j**) October, (**k**) November, (**l**) December, and (**m**) annually. ($p < 0.05$).

Similarly, a pattern of LST trends was found for the TB at an annual scale (Figure 9m). The LST rate was between −0.4 °C/10 a and 0.5 °C/10 a. The desert was dominated by a decreasing trend, while the surrounding regions were dominated by an increasing trend.

Figure 10 shows the LST trends for different seasons. In spring, the annual LST rate was between −0.4 °C/10 a and 0.6 °C/10 a. The lowest rate (−0.4 °C/10 a) was observed in the northern, northeastern, and southern parts of the basin; the highest rate (0.6 °C/10 a) was evident in the eastern southwestern part of the basin. Most of the study region was dominated by an increasing trend. In summer, the annual LST rate was between −0.6 °C/10 a and 0.5 °C/10 a. The oasis regions were dominated by a decreasing trend, and the desert was dominated by an increasing trend. The majority of the study area was dominated by an increasing trend. In autumn, the annual LST rate was between −0.3 °C/10 a and 0.7 °C/10 a. The southern regions were dominated by an increasing trend, and the northern regions were dominated by a decreasing trend. The majority of the study area was dominated by an increasing trend. In winter, the annual LST rate was between −0.5 °C/10 a and 0.6 °C/10 a. The eastern regions were dominated by

a decreasing trend, while the western regions were dominated by an increasing trend. The majority of the study area was dominated by a decreasing trend.

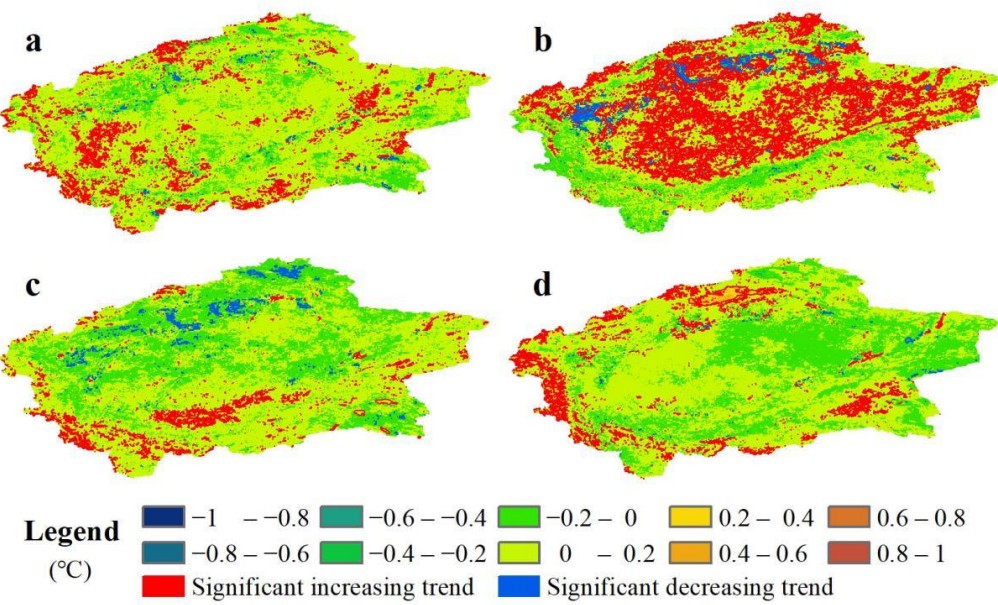

**Figure 10.** Land surface temperature (LST) trends in the Tarim Basin (TB) for spring (**a**), summer (**b**), autumn (**c**), and winter (**d**). ($p < 0.05$).

### 3.5. Importance of Natural and Perceived Factors

Major factors affecting LST are latitude, solar radiation, topography, and the nature of the underlying surface. Eight indicators (i.e., mean air temperature, maximum air temperature, minimum air temperature, air pressure, relative humidity, wind speed, precipitation, and elevation) were applied to study the principal factors influencing LST changes. The RF model was used to analyze the selected indicators quantitatively, and the results are shown in Figure 11a. Elevation was the most significant influencing factor, explaining 22.1% of the variation in LST. In Figure 11b, the correlation coefficients of mean, maximum, and minimum air temperatures were high, at 0.99, 0.99, and 0.98, respectively, with significant positive correlations. Relative humidity and elevation had an obvious negative correlation, with correlation coefficients of −0.5 and −0.1, respectively, indicating that air temperature, relative humidity, and elevation had a good correlation with LST.

### 3.6. Impact of Land Use/Cover Type on LST

The spatio-temporal distribution of LST is mainly affected by factors such as latitude, solar radiation, topography, altitude, and the nature of the underlying surface. The TD is surrounded by high mountains in the north, west, and south; therefore, altitude and land use/cover types had a large impact.

Figure 12 shows the LST over regions with different land-use types in different months. The results show that during February–December, the lowest LST was observed over permanent glacier snow due to factors such as altitude, while the highest LST was observed in the desert due to factors such as air dryness, a lack of vegetation, and low latitude. In January and December, the lowest LSTs were observed over permanent glacier snow (−7.8 °C and −23.4 °C, respectively) and the highest LSTs were observed over reservoir ponds (7.4 °C and −7.9 °C, respectively).

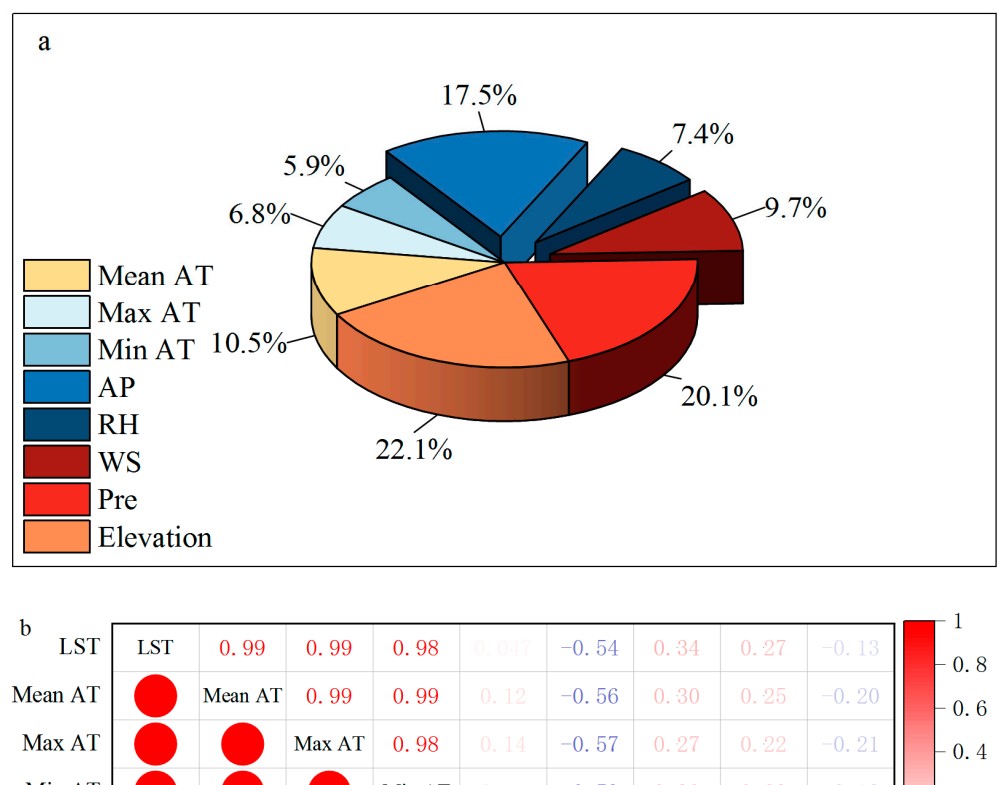

**Figure 11.** (**a**) Random forest (RF) model simulation of land surface temperature (LST) and (**b**) correlation between LST and influencing factors in the Tarim Basin (TB).

Construction and dry land were easily affected by human activities. Sparse grassland had minimal vegetation coverage, and the temperature increased rapidly with solar radiation; therefore, the LST was relatively high. On the one hand, because different land use/cover types had different characteristics, vegetated areas such as grasslands promoted energy exchange between the ground surface and the near-ground atmosphere, leading to a low LST; on the other hand, climatic conditions (e.g., topography, sunshine, and precipitation) of areas with the same land use type could also be different. In general, the annual and monthly average LST variations of different land use/cover types were consistent, and an increase in vegetation cover caused LST to be lower.

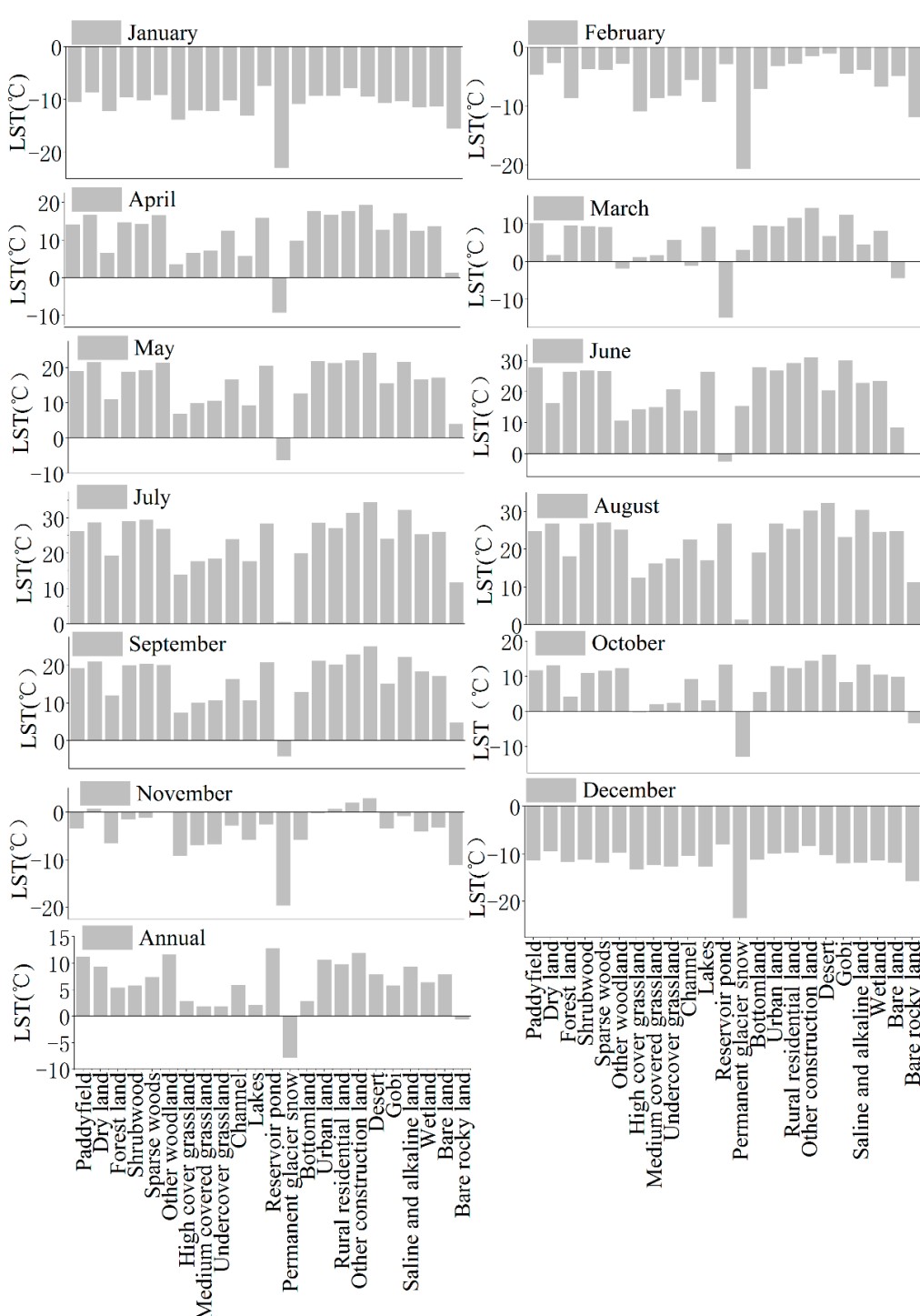

**Figure 12.** Monthly variation of land surface temperature (LST) for different land-use types in the Tarim Basin (TB).

### 3.7. Influence of Atmospheric Circulation on LST

The TB is located far from the ocean; however, the region may be considerably impacted by weather changes over the sea. We calculated the LST anomaly and analyzed the consistency of the LST anomaly with three indices: the sea surface temperature (SST) index of the NINO3.4 region (Figure 13a), the NAO index, and the PDO index. In the past 19 years, six El Nino events have occurred, during August 2002–February 2003, August 2004–January 2005, September 2006–January 2007, July 2009–April 2010, May 2015–April 2016, and October 2018–June 2019. Among them, the El Nino event in 2015–2016 was

the strongest and continued for 1 year, and the maximum NINO3.4 index reached 2.6 °C. In the past 19 years, four La Nina events occurred, during August 2007–May 2008, June 2010–May 2011, August 2011–February 2012, and September 2017–March 2018. The LST in the TB showed a periodic maximum during each El Nino event, whereas every La Nina event caused a periodic minimum LST; this indicates that both El Nino and La Nina had a strong influence on the LST variations in the TB. The LST reached the highest level in 2017 under the influence of the strong El Nino event in 2015 and then decreased over the following two years. There is consistency between the NAO (Figure 13b) and PDO (Figure 13c) indices and the mean LST anomaly, indicating that these strongly influenced the variation in LST in the TB.

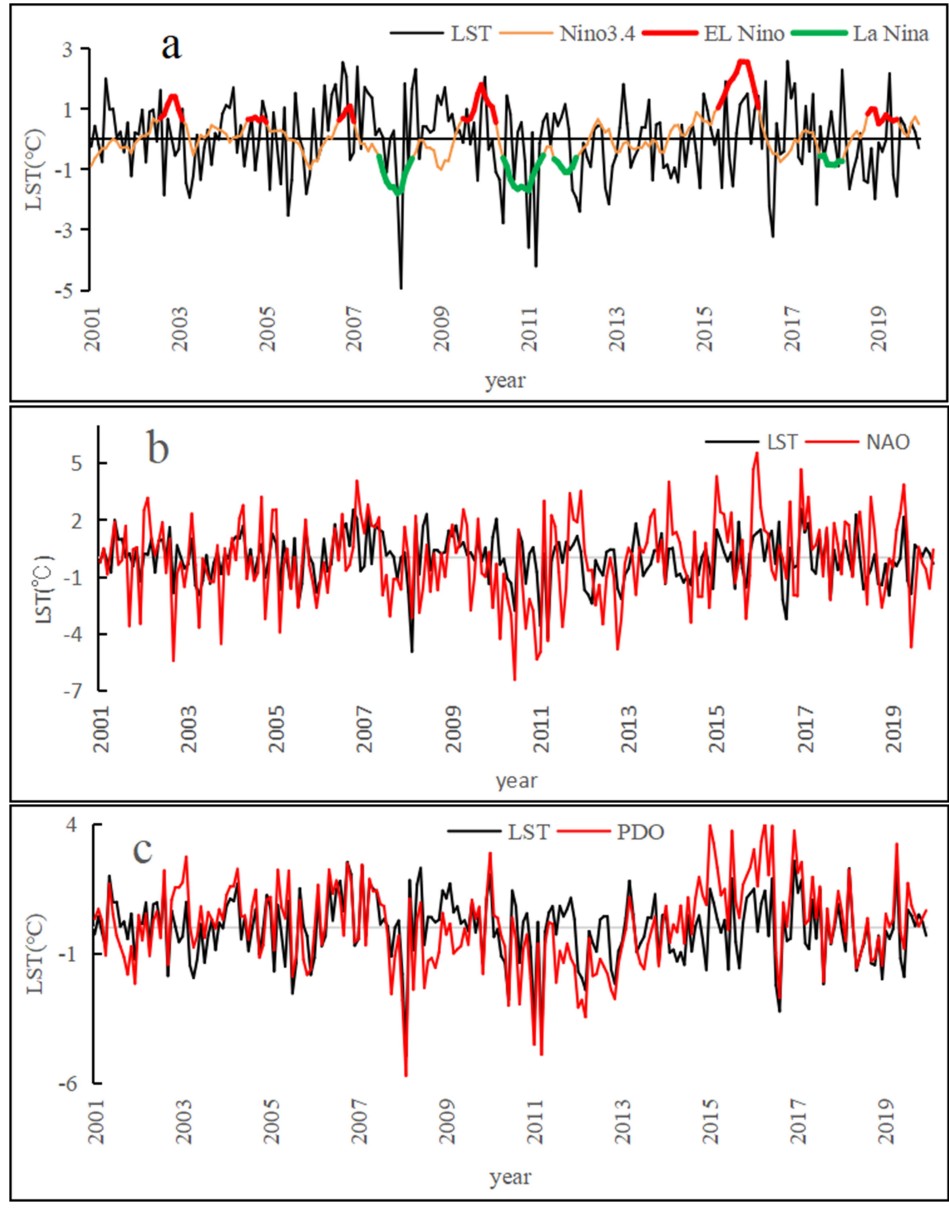

**Figure 13.** Variations in the land surface temperature (LST) anomalies and three climate indices. (**a**) LST anomaly in the Tarim Basin (TB) and the sea-surface temperature (SST) anomaly in the NINO3.4 area during 2001–2019. (**b**) Monthly LST anomaly in the TB and the North Atlantic Oscillation (NAO) index. (**c**) Monthly LST anomaly in the TB and the Pacific Decadal Oscillation (PDO) index.



## 4. Discussion

LST is a significant indicator of the thermal environment of a region. Previous studies mainly researched the effects of single elements on LST, such as the NDVI [39], topography [40], and ozone pollution [41]. Some researchers have carried out relevant investigations in the TB using Landsat [42–44], MODIS [45], and ground observation [46] data. However, there is a lack of large-scale and systematic research on the entire region. LST changes are affected by climate change, land use, human activity, and hydrographic factors in the TB, and these changes are mainly caused by solar radiation and underground conditions. LST land based observations are essential for studying thermal conditions; however, many high-altitude and desert areas remain heavily under-sampled. To overcome those limitations, this study assessed and detected LST changes using multi-temporal MODIS satellite data. Air temperature, relative humidity, wind speed, air pressure, precipitation, elevation, and land-cover type were used to analyze the relationships between LST and the influencing factors. The results show that the main climatic factors affecting the actual LST in the TB were air temperature, relative humidity, and elevation, and that precipitation, air pressure, and wind speed had minimal effects on the interannual and seasonal variations of LST.

Research results have shown that topography has a great influence on the LST [47–50]. This was especially prominent in the northern, western, and southern regions of the TB, where relatively large elevation differences and steep gradients bring about strong spatial LST gradients. On the contrary, desert regions, where the topography was more even, showed lower intra-regional LST gradients. In addition to elevation land use/land cover also plays a significant role in explaining spatio-temporal LST patterns. Vegetation provides shade and assists in reducing the LST, which reduces incident radiation and evapotranspiration, which helps to regulate overheating [51]. The results of our study agree with research results by Adeyeri et al. [52] which shows that the lowest LST was observed in water bodies. Research results show that an urban area with green vegetation suffers less from LST than a non-vegetated urban area [53]. Urban lands feature higher temperatures than their natural counterparts and the intra-annual gradients are different amongst the different land uses [54].

Human activities also affected the LST distribution. As the population increases, the vegetation resources and water of a watershed begin to be used intensively. Hence, farmlands increase LST [55]. In addition, there is a high correlation between LST and GDP [56]. Vegetation coverage changes with an increase in the farmland area. Desert and sandy land have little or no vegetation. In the present study, an increase in NDVI led to a lower albedo, and thus relatively higher vegetation coverage rates were associated with a lower albedo. These results suggest a negative correlation between vegetation coverage and albedo. Specifically, albedo declines with increasing vegetation coverage. During photosynthesis and transpiration, green plants absorb heat and $CO_2$ from the air and thus, lower the atmospheric temperature.

Studies showed that the climate changes in Asia might be related to the Arctic Oscillation [57], and Asia-Pacific Oscillation [58]. In recent years, studies have been conducted to figure out the relationship between LST and atmospheric circulations [59,60], LST changes in deserts may be modulated by Earth system variables such as ocean surface circulations, surface longwave radiation, and atmosphere teleconnections [61,62]. Studies have shown that there is a certain relationship between the air temperature in TB and ENSO [63,64]. Further investigation results showed that both ENSO and PDO affect the air temperature of the TB [65]. However, the relationship between the ENSO phenomenon and the LST of the TB is not clear. This research conducted a detailed analysis of this issue and obtained corresponding conclusions.

Studying the LST change and influencing factors is providing a theoretical basis for the decision-making of governmental authorities and may provide the basis for understanding regional thermal conditions, urbanization, atmospheric circulation, and local circulation in oasis and desert.

## 5. Conclusions

In the study period, high LSTs occurred in the desert and plains of the TB, while low LSTs occurred in surrounding mountain regions. The spatial distributions of annual average LST in different months varied; the highest LST occurred in July (25.1 °C), while the lowest LST was observed in January (−9.5 °C). On a seasonal scale, LST decreased in the order: summer > spring > autumn > winter. There was a different variation trend in the monthly LST of the entire study area, whereby the annual LST showed an increasing trend with a rate of 0.2 °C/10 a. The desert and surrounding plains were dominated by a decreasing trend, and the mountain area was dominated by an increasing trend. LSTs in April and July were dominated by an increasing trend, while those in May and November were dominated by a decreasing trend. In different seasons, the LST trends ranged from −0.2 °C/10 a to 0.2 °C/10 a. In spring and autumn, western regions were dominated by a decreasing trend, whereas eastern regions were dominated by a decreasing trend in winter. In summer, areas covered by vegetation were dominated by a decreasing trend, and desert and bare lands were dominated by an increasing trend.

Through RF model analysis, the influence of various factors on LST in the TB was clarified; elevation was the most significant influencing factor (22.1%), followed by mean air temperature (20.1%). Other influencing factors directly or indirectly affected the LST. The correlation analysis indicated that the main climatic factors air temperature, relative humidity, and elevation have a good correlation with the LST in the TB and that the precipitation, air pressure, and wind speed had little correlation with the interannual and seasonal variations of LST.

Land-cover type also affected LST. The results show that during February–December the lowest LST occurred over permanent glacier snow, while the highest LST occurred in the desert. In January and December, the lowest LSTs were observed over permanent glacier snow (−7.8 °C and −23.4 °C, respectively) while the highest LSTs were observed over reservoir ponds (7.4 °C and −7.9 °C, respectively).

El Nino and La Nina events greatly influenced the variation in LST in the TB. The NAO and PDO indices were consistent with the mean LST anomaly, indicating that these greatly influenced the variations in LST in the TB.

**Author Contributions:** Methodology, A.A.; supervision, B.L., Q.H. and J.L.; writing—original draft, A.A.; writing—review and editing, J.L., A.S. and Y.Y. Conceptualization—L.J. All authors have read and agreed to the published version of the manuscript.

**Funding:** This work was financially supported by the National Natural Science Foundation of China (42030612, 41830968) and the Second Tibetan Plateau Scientific Expedition and Research (STEP) program (Grant No. 2019QZKK010206).

**Institutional Review Board Statement:** Not applicable.

**Informed Consent Statement:** Not applicable.

**Data Availability Statement:** Not applicable.

**Acknowledgments:** We would like to thank the Xinjiang Meteorological Administration (XMA) for providing the meteorological data. All individuals included in this section have consented to the acknowledgement.

**Conflicts of Interest:** The authors declare no conflict of interest.

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
