# Peer review of "Spatio-Temporal Changes of Land Surface Temperature and the Influencing Factors in the Tarim Basin, Northwest China"

_remotesensing, doi:10.3390/rs13193792_

Round 1
Reviewer 1 Report
Dear all,
Thank you for the opportunity to read this interesting paper.
I think the study is scientifically interesting and the work is well structured. Results are thoroughly presented. Only the below aspects should be addressed by the author before publication.
SPECIFIC COMMENTS:
- Line 99: Why the range considered is 1960-2011 and not 1960-2019? It would be interesting to see also in the last years of the analysed
-Line 117: At what height are the data extracted? Height of the control units?
- Discussion: Maybe this section could be further developed.
Author Response
Response to reviewers’ comments on Remote Sensing-1324032
Response to Reviewer 1 comments:
- Line 99:Why the range considered is 1960-2011 and not 1960-2019? It would be interesting to see also in the last years of the
Response: Thanks for your remark. In my original analysis, i have quoted the references (Zhou, C.; Wang, K. Land surface temperature over global deserts: Means, variability, and trends. J. Geophys. Res. Atmos. 2016, 121,14344–14357.) but in this revised manuscript, i have used data based on ground observation from 1961 to 2019 which obtained from the Meteorological Bureau of the Autonomous Region
- Line 117:At what height are the data extracted? Height of the control units?
Response: Thanks for your remark. We used ground-based monthly maximum air temperature (Max AT), minimum air temperature (1.5 m ± 5 cm) (Min AT), mean air temperature (1.5 m) (Mean AT), air pressure (1.2 m) (AP), relative humidity (1.5 m) (RH), wind speed (10.8 m ± 5 °) (WS), precipitation (0.7 m ± 3 cm) (Pre), and elevation data from 39 weather stations during 2001‒2019.
- Line 117:Discussion: Maybe this section could be further developed.
Response: Thanks for your remark. I have further developed the discussion part, and added the followings sentences:
Line 472-477: LST is an significant indicator of the thermal environment of a region. Previous studies mainly researched the affects of single elements on LST, such as the NDVI [39] ,topography [40] , and ozone pollution [41]. Some researchers have carried out relevant investigtions in the TB using Landsat [42,44], MODIS [45],and ground observation [46] data. However, there is a lack of large-scale and systematic research on the entire region. LST changes are affected by climate change, land use, human activity.
Line 492-500: In addition to elevation land use/land cover also plays an significant role in explaining spatio-temporal LST pattern. Vegetation provides shade and assists in reducing the LST, which reduces incident radiation and evapotranspiration, which helps to regulate overheating [51]. Results of our study agree with research results by Adeyeri et al [52] which showes that the lowest LST was observed in water bodies. Research results show that an urban area with green vegetation suffers less from LST than a non-vegetated urban area [53]. Urban lands feature higher temperatures than their natural counterparts and the intra-annual gradients are different amongst the different land uses [54].
Line 511-524: Studies showed that the climate changes in Asia might be related to the Arctic Oscillation [57], and Asia-Pacific Oscillation [58]. In recent years, studies have been conducted to figure out relationship between LST and atmospheric circulations[59,60], LST changes in deserts may be modulated by Earth system variables such as ocean surface circulations, surface longwave radiation, and atmosphere teleconnections [61,62]. Studies have shown that there is a certain relationship between the air temperature in the TB and ENSO [63,64]. Further investigation results showed that both ENSO and PDO affects on air temperature of the TB[65]. However, the relationship between ENSO phenomenon and the surface temperature of the Tarim Basin is not clear.This research conducted a detailed analysis of this issue and obtained corresponding conclusions.
Studing the LST change and influencing factors is providing theoretical basis for the decision-making of governmental authorities and may provides basis for understand regional thermal conditions, urbanization, atmospheric circulation, local circulation in oasis and desert.

Reviewer 2 Report
The manuscript entitled "Spatio-temporal Changes of Land Surface Temperature in the Tarim Basin and the Influencing Factors" has disadvantages, which should be corrected in order for this manuscript to become a scientific paper. Only some characteristic disadvantages (D) will be mentioned here, which should guide the authors in preparing a new revised version of the manuscript.
D1.) The word "China" should be added to the title of the manuscript, so that it is immediately clear to which country the study area belongs.
D2.) Lines 14 (abstract) and 38. Need to define what "Land surface temperature (LST)" is? Is it the same as "surface air temperature" (World Meteorological Organization (WMO), Technical Regulations, No. 2) or is it some other temperature?
D3.) Lines 21 - 23 (abstract), and 55. Instead of the text:
"... (25.13 °C) and the lowest was in January (-9.49 °C). On a seasonal scale, LST decreased in the order: summer > spring > autumn > winter. Annual LST showed an increasing trend of 0.23 °C/decade in...",
should be written:
"... (25.1 °C) and the lowest was in January (–9.5 °C). On a seasonal scale, LST decreased in the order: summer > spring > autumn > winter. Annual LST showed an increasing trend of 0.2 °C/decade in...".
Note 1: The error of the calculated temperature (0.01 ºC) cannot be less than the error obtained by direct measurement of temperature (0.1 ºC).
Note 2: The temperature should only be displayed to one decimal place (see WMO, Technical Regulations). This applies to all places in the manuscript where the temperature is displayed to more than one decimal place. Everything needs to be fixed.
Note 3: The long dash "–" is the mathematical sign for "minus" and the short dash "-" is the notation for the rest applications.
D4.) For example, lines 93 and 99. Instead of the text:
"... 800 to 1,300 m... 10 to 15 °C…",
should be written:
"... 800 m to 1,300 m... 10 °C to 15 °C...".
Note 4: Each number representing a physical quantity must have a unit unless an interval is displayed. Everything needs to be fixed.
D5.) Figure 1. The location of the Study area should be more clearly shown on maps of China and/or Asia.
D6.) Line 105. Instead of the text "Location of the study area." should be written, “The territory of the study area, its locations in China and/or Asia, distribution of meteorological stations, and landscape”.
D7.) Lines 108 - 136 (2.2 Data). In general, the data are very poorly presented, especially the following:
D7.1.) Lines 109 - 115 (2.2.1 MODIS LST Data). It is necessary to write what kind of data this is, what values they represent, in which spatial-temporal network they are given, how often they are, what is software and/or way they are downloaded from the database, etc.
D7.2.) Lines 117 - 122 (2.2.2 Meteorological Data). It is necessary to write to which observation program according to the WMO classification the meteorological data belong (synoptic observation program, climatological observation program, or something else). It is necessary to state from which meteorological stations the data originate and give the units in which the data were used, to describe the time series of each of the listed types (daily, monthly or annual series), the lengths of the time series with the number of data in each series. The source of the data is given, but without a link or some other link that information is useless, so I suggest that the data source be better described.
Example 1: Below I enclose a reference as an example of how the data should be described. Only temperature data is described in this reference, so other data used in the manuscript should be described analogously.
Gavrilov, M. B., Tošić, I., Marković, S. B., Unkašević, M., Petrović, P., 2016: The analysis of annual and seasonal temperature trends using the Mann-Kendall test in Vojvodina, Serbia. Idöjárás (Quarterly Journal of the Hungarian Meteorological Service), 120-2, 183-198.
D7.3.) Lines 124 - 126 (2.2.3 Climate Index Data). Abbreviations should be defined: NINO3.4, NAO, and PDO, or appropriate references should be given. Also, it is necessary to show what this data represents and state how it was taken from the mentioned databases, automatically by using some software or in another way.
D8.) Lines 148 - 152. Text "Eight concomitant variables, namely mean air temperature (Mean AT), maximum air temperature (Max AT), minimum air temperature (Min AT), air pressure (AP), relative humidity (RH), wind speed (WS), precipitation (P), and elevation (E) were considered. Within the RF model framework, concomitant variables were evaluated and the significance of concomitant variables was obtained." belongs to the data description and should be moved to chapter 2.2.2 Meteorological Data.
D9.) Lines 263 – 338 (3.4 LST Trends). Regression analysis is not enough to reliably establish a trend sign, which is done in this text. In order to establish the trend sign more reliably, a trend assessment should be applied. The Mann-Kendall (MK) test is often used for this purpose. I recommend the authors use the MK test to check their claims about the trend sign in all cases. For this purpose, the already mentioned reference can serve as an example.
D10.) Continuing critical reading of the manuscript is impossible without accepting corrections from D1 to D9. That's why I stopped reading here, and I can continue it in the second iteration if it happens, and the final assessment of the scientific value of this manuscript will be made after the major revision.
Author Response
Response to reviewers’ comments on Remote Sensing-1324032
Response to Reviewer 2 comments:
- The word "China" should be added to the title of the manuscript, so that it is immediately clear to which country the study area belongs.
Response: Thanks for your remark. I have changed the original title (Spatio-temporal Changes of Land Surface Temperature in the Tarim Basin and the Influencing Factors) to (Spatio-temporal Changes of Land Surface Temperature and the Influencing Factors in the Tarim Basin,northwest China).
- Lines 14(abstract) and 38. Need to define what "Land surface temperature (LST)" is? Is it the same as "surface air temperature" (World Meteorological Organization (WMO), Technical Regulations, No. 2) or is it some other temperature?
Response: Thanks for your remark. I have defined the "Land surface temperature (LST)" as Land surface temperature (LST) refers to the temperature of soil, water, buildings and vegetation canopy on the land surface [1].
It is not the "surface air temperature", it is indicates the air temperature (1.5 m).
"... (25.13 °C) and the lowest was in January (-9.49 °C). On a seasonal scale, LST decreased in the order: summer > spring > autumn > winter. Annual LST showed an increasing trend of 0.23 °C/decade in...",
should be written:
"... (25.1 °C) and the lowest was in January (–9.5 °C). On a seasonal scale, LST decreased in the order: summer > spring > autumn > winter. Annual LST showed an increasing trend of 0.2 °C/decade in...".
Note 1: The error of the calculated temperature (0.01 ºC) cannot be less than the error obtained by direct measurement of temperature (0.1 ºC).
Note 2: The temperature should only be displayed to one decimal place (see WMO, Technical Regulations). This applies to all places in the manuscript where the temperature is displayed to more than one decimal place. Everything needs to be fixed.
Note 3: The long dash "–" is the mathematical sign for "minus" and the short dash "-" is the notation for the rest applications.
"... 800 to 1,300 m... 10 to 15 °C…",
should be written:
"... 800 m to 1,300 m... 10 °C to 15 °C...".
Note 4: Each number representing a physical quantity must have a unit unless an interval is displayed. Everything needs to be fixed.
Response: Thank you very much for your nice remarks. I have changed the whole places according to your comments.
Response: Thank you very much for your nice remarks. I have fixed the figure 1 according to your comments.
Response: We are thankful for your kind hint. I have addedd (The territory of the study area.Located in northwestern China, lies between Tianshan Mountain, Kunlun and Altun Mountain.The black, green, and red circles represent stations in oases, desert, and mountainous terrain, respectively) under the figure.
Response: Thank you very much for your nice remarks. I have added (Line 114-119: In this study, the MODIS/Terra and Aqua LST/Emissivity Daily L3 Global 1 km SIN Grid product data is used and downloaded from the NASA EarthData Search (https://developers.google.com/earth-engine/datasets/catalog/MODIS_006_MOD11A1). Both MODIS/Terra and Aqua pass over the Tarim Basin twice a day. The MOD11A1 product are uses MODIS bands 31 and 32 which produced by the generalized split-window algorithm [20] (10.78–11.28 µm and 11.77–12.27 µm, respectively). in this pace.
Example 1: Below I enclose a reference as an example of how the data should be described. Only temperature data is described in this reference, so other data used in the manuscript should be described analogously.
Gavrilov, M. B., Tošić, I., Marković, S. B., Unkašević, M., Petrović, P., 2016: The analysis of annual and seasonal temperature trends using the Mann-Kendall test in Vojvodina, Serbia. Idöjárás (Quarterly Journal of the Hungarian Meteorological Service), 120-2, 183-198.
Response: Thank you very much for your nice remarks. I have added (Line 129-146: We used ground-based monthly maximum air temperature (Max AT), minimum air temperature (1.5 m ± 5 cm) (Min AT), mean air temperature (1.5 m) (Mean AT), air pressure (1.2 m) (AP), relative humidity (1.5 m) (RH), wind speed (10.8 m ± 5 °) (WS), precipitation (0.7 m ± 3 cm) (Pre), and elevation data from 39 weather stations during 2001‒2019. A total of 4,30,and 5 meteorological stations were selected to represent mountainous areas,oases, and desert areas,respectively [24]. For the verification of MODIS LST data in this sdudy usded the monthly LST situ-observation data (The sensing part and the surface body are half buried in the soil) from 39 weather stations during 2001-2019. This study also used the annual average air temperature and precipitation data of 39 meteorological stations in the TB from 1961 to 2019 in order to introduce the characteristics of climate change of the study area. The locations of stations are presented in Figure 1. Meteorological data belong synoptic observation program and each series contains 8892 data (result part) and 59 data (study area part). The data were provided by the Xinjiang Meteorological Administration and underwent strict quality control prior to being released.
Eight concomitant variables, Mean AT, Max AT, Min AT, AP, RH, WS, Pre, and elevation were considered. Within the RF model framework, concomitant variables were evaluated and the significance of concomitant variables was obtained.
D7.3.) Lines 124 - 126 (2.2.3 Climate Index Data). Abbreviations should be defined: NINO3.4, NAO, and PDO, or appropriate references should be given. Also, it is necessary to show what this data represents and state how it was taken from the mentioned databases, automatically by using some software or in another way.
Response: Thank you very much for your nice remarks. I have added (Line 149-155: NINO3.4 is the exceed 0.4°C for 6 months running mean sea surface temperature anomaly in the region (5°S ~ 5°N, 120°W ~ 170°W) and has large variability on El Niño time scales [25]. The North Atlantic Oscillation(NAO)index is the indicator of the NAO, and is the difference of normalized mean zonal sea level pressure between the Azores and Iceland [26]. The Pacifc Decadal Oscillation (PDO) is a Decadal cycle of climate change in the Pacifc ocean and characterized by unusually warm or cold surface water temperatures in areas north of 20°N in the Pacifc Ocean [27].
Response: We are thankful for your kind hint. I have used the Sen's-slop and Mann-Kendall (MK) test to analyses the trend and significance level of these data.

Round 2
Reviewer 2 Report
The authors seem to have improved the revised manuscript (remotesensing-1324032-peer-review-v2), although I have some other objections, for example:
- why did they increase the number of decimal places for some numbers? It would be better if they reduced the number of decimal places and reduced them to the rank of measurement error;
- all physical values do not have units that reduce clarity in reading the text; and
- some physical values are displayed with an error, but the error is displayed as a percentage, and it is better that error to be in the same unit as the value being displayed.
I suggest that the manuscript be accepted for publication, but it would be better to accept the previous remarks beforehand, in order to make the manuscript more readable.
Author Response
Response to reviewers’ comments on Remote Sensing-1324032
Response to Reviewer 2 comments:
- The authors seem to have improved the revised manuscript (remotesensing-1324032-peer-review-v2), although I have some other objections, for example:- why did they increase the number of decimal places for some numbers? It would be better if they reduced the number of decimal places and reduced them to the rank of measurement error;- all physical values do not have units that reduce clarity in reading the text; and- some physical values are displayed with an error, but the error is displayed as a percentage, and it is better that error to be in the same unit as the value being displayed.Line 24. I have changed the (decade) to (10 a).Line 58. I have changed the (decade) to (10 a).Line 103. I have added the (mm) in front of the to.Line 133. I have deleted ± 5 °. Line 243. I have added the (°C) in front of the to.Line 261. I have changed the color of ( –) from red to black.Line 278. I have added the (°C) in front of the to.Line 301. I have added the (°C) in front of the to.Line 322. I have added the (°C/10 a) in front of the and.Line 329. I have added the (°C/10 a) in front of the and.Line 334. I have added the (°C/10 a) in front of the and.Line 342. I have changed the format of (°C).Line 350. I have added the (°C/10 a) in front of the and.Line 355. I have added the (°C/10 a) in front of the to.Line 357. I have added the (°C/10 a) in front of the and.Line 360. I have added the (°C/10 a) in front of the and.Line 365. I have added the (°C/10 a) in front of the and.Line 379. I have added the (°C/10 a) in front of the and.Line 385. I have added the (°C/10 a) in front of the and.Line 421. I have changed the format of (°C).Line 530. I have added the (°C/10 a) in front of the to.Line 545. I have added the (°C) in front of the and.
- Line 544. I have added the (°C) in front of the and.
- Line 422. I have changed the format of (°C).
- Line 403. I have changed the color of ( –) from red to black.
- Line 382. I have added the (°C/10 a) in front of the and.
- Line 375. I have added the (°C/10 a) in front of the and.
- Line 363. I have added the (°C/10 a) in front of the to.
- Line 359. I have added the (°C/10 a) in front of the to.
- Line 356. I have added the (°C/10 a) in front of the and.
- Line 352. I have added the (°C/10 a) in front of the and.
- Line 345. I have added the (°C/10 a) in front of the and.
- Line 338. I have added the (°C/10 a) in front of the and.
- Line 330. I have changed the format of (°C).
- Line 325. I have added the (°C/10 a) in front of the and.
- Line 319. I have added the (°C/10 a) in front of the and.
- Line 300. I have added the (°C) in front of the to.
- Line 273. I have added the (°C) in front of the to.
- Line 255. I have changed the color of ( –) from red to black.
- Line 244. I have changed the (decade) to (10 a).
- Line 104. I have changed the (decade) to (10 a).
- Line 102. I have changed the (decade) to (10 a).
- Line 55. I have changed the (decade) to (10 a).
- Response: Thanks for your remark. Some decimal places increased may due to using different method. I have added all units that last time missed.
